# Environmental Factors and Genetic Parameters of Beef Traits in Fleckvieh Cattle Using Field and Station Testing

**DOI:** 10.3390/ani10112159

**Published:** 2020-11-19

**Authors:** Radek Filipčík, Daniel Falta, Tomáš Kopec, Gustav Chládek, Milan Večeřa, Zuzana Rečková

**Affiliations:** Faculty of AgriSciences (FA), Mendel University in Brno, Zemědělská 1, 613 00 Brno, Czech Republic; radek.filipcik@mendelu.cz (R.F.); tomas.kopec@mendelu.cz (T.K.); gustav.chladek@mendelu.cz (G.C.); milan.vecera@mendelu.cz (M.V.); zuzana.reckova@mendelu.cz (Z.R.)

**Keywords:** Czech Fleckvieh, beef traits, fatness, SEUROP, cattle, heritability

## Abstract

**Simple Summary:**

Meat production plays an important role in the efficiency of rearing dual-purpose breeds of cattle in Europe. It is generally known and accepted that most of the beef produced in the EU is produced from dairy herds. This fact provided the motivation for analyzing the influence of genetic and environmental factors on the indicators of meat production from Fleckvieh (known also as dairy Simmental) cattle in the Czech Republic and the interrelationships between these factors. The analysis included evaluating the statistical relationship between the degree of fatness, the heritability and genetic correlation with other traits and also the relationship with exterior muscularity. Together with this analysis, the normal indicators, which are usually evaluated in the population, were also analyzed. The results of our study indicate that meat yield traits are positively related to the live weight of the animal. These findings create possibilities for application in breeding strategies of dual-purpose cattle.

**Abstract:**

The goal of this study was to analyze the genetic and environmental factors of selected meat yield indicators in Fleckvieh cattle in the Czech Republic, through the application of station (S) and field (F) testing methods. Data collected from fattened bulls were analyzed for F (*n* = 9378) and for S (*n* = 6346). In the F method and the S method, the values of the main meat yield indicators were as follows: carcass weight 402.91 kg (F), 339.37 kg (S); carcass daily gain 626.05 g/day (F), 609.74 g/day (S); SEUROP carcass classification 2.73 (F), 2.19 (S). Environmental factors were found to have a significant impact on the selected meat yield indicators; their heritability ranged from 0.14 (SEUROP classification) to 0.33 (dressing percentage). The genetic trend was significantly positive only in relation to those meat yield traits, which had a positive link to the size or weight of the animal. The genetic correlation between observations obtained in the S and F methods of testing was very high in relation to the carcass daily gain (0.8351) and carcass weight (0.8244), while slightly lower correlations were calculated for the SEUROP classification. A genetic evaluation of the degree of fatness is not routinely performed in Fleckvieh populations, and the newly established heritability for this trait ranges between 0.17–0.20. The genetic correlation between beef yield indicators and the exterior trait of muscularity was also established, and shows a strong link to the net daily gain, the SEUROP classification and body weight (0.79–0.97). The aim of the study was to evaluate the genetic and environmental effects on meat yield and also estimate genetic parameters for new traits. We can also state, based on the results, that a strong positive genetic trend is confirmed, especially in traits related to the size or weight of animals. This result can be used in breeding programs of dual-purpose cattle, where we can genetically improve the meat and milk yield through the body size.

## 1. Introduction

The meat yield testing in the station method (S) and the field method (F) was analysed by [1,2]. Analysis of meat yield from Fleckvieh cattle at fattening control station (FCS) stations was pursued by [3] and provided important guidance on analyses for meat traits in this breed. Fleckvieh is the fourth biggest population of dual-purpose cattle in the world and the focus of genetic improvement has been widely put on milk and fitness traits in recent years [4,5,6]. Genetic improvement for meat yield in dual-purpose cattle breeds was previously negatively affected by the lack of information for estimating meat yield breeding values. International genetic evaluation studies of meat yield at Fleckvieh cattle was implemented in Germany, Austria, the Czech Republic and Hungary in 2007. Data from station (method S) and field testing (method F) are registered in the system and are available.

The overall meat yield evaluation consists of a composite of traits in cattle, including body weight gain, expressed as the increase in live weight (kg per day) and also the increase in carcass weight (kg per day). Body weight, or carcass weight, and the derived dressing percentage are other significant quantitative meat yield parameters included in the analysis. Fleshiness ratings according to the SEUROP classification system, along with fatness and exterior muscularity rating values, also play a significant role in the qualitative expression of the meat yield. These and other meat yield indicators and carcass composition rating values were analyzed in the Fleckvieh breed by Chládek et al. [7,8] and other authors [3,9,10]. A number of authors [11,12,13,14] pursued the links between meat yield indicators, particularly genetic correlation, in great detail. For example, [15] discussed the estimates of genetic parameters in Fleckvieh. The influence of various environmental factors and the type of fattening system (organic farming for instance) was discussed by [16].

The environmental (non-genetic) and genetic factors affecting important meat yield traits in fattened bulls of the Fleckvieh breed were analyzed in this study. The genetic parameters and correlation between the main meat yield indicators were estimated. The source of information was taken into consideration, i.e., whether the data about meat yield was collected at specialized fattening control stations (FCS), and fattened here under standardized conditions and subsequently slaughtered (station test—method S), or whether it was obtained directly from the commercial stock (population), from data about the commercial slaughter of bulls at slaughterhouses in the Czech Republic (field test—method F).

## 2. Materials and Methods

### 2.1. Dataset

Various types of data routinely collected at FCS (S method) and commercial slaughterhouses (F method) were used for the purpose of this study. Thus, no animals were handled directly during our research, nor were any animals subjected to invasive methods, which is why approval by the Ethics Committee was not required or obtained. Data about the slaughter of bulls from station tests and from field tests, i.e., from slaughterhouses, were used to calculate and estimate genetic parameters and breeding values and assess environmental factors affecting meat yield. All the data, commencing from the year of the birth of the bulls in 2004 and later, were used from the station test. This included data from 6346 bulls (Table 1). A much bigger number of animals can be used in relation to data from the field test, but the whole file of data from the field test was reduced to 9378 bulls (according to the birth year and size of contemporary group) for the purpose of comparison with station data and also due to the numerical demands of REML(restricted maximum likelihood) during multi-trait analysis. The descriptive statistics and the frequency of individual explanatory variables used during linear modelling are given in Table 1 and Table 2. This means that two types of data entered the calculation: data from station tests (method S), provided by the observation of the in situ cattle fattening ability (FCS—fattening control station), and data from slaughterhouses (field test—method F). The FCS were used to collect data about meat yield for routine estimates of breeding values. The male progeny of bulls used for insemination (at least 12 bull calves of the same age from a bull used for insemination) were obtained for the station. Fattening took place at two stations, using identical feed formulations and quantities for fattening (in three stations until 2011) up to 610 days of age. The animals were then slaughtered, and their body weight (BW), carcass weight (CW), SEUROP (EUR) classification, fatness (FAT), meat yield (DP), net gain (CDG) and also muscularity rating (MUS) before slaughter were established.

Data collected from F were on slaughtered bulls from farms in the Czech Republic, and therefore, are obtained from uncontrolled field tests. Data on a smaller number of traits obtained here are compared to the station test, specifically carcass weight (CW), net gain (CDG), SEUROP (EUR) classification and fatness (FAT). The CDG is the ratio of carcass weight to age of animal at slaughter (g/day). Carcass is defined as the muscle, bone and fat associated with the slaughter of an animal, left after the removal of the head, hide, hooves and internal organs. DP is the percentage of the weight of the carcass from live body weight before slaughtering. The SEUROP classification and degree of fatness from both sources of data are based on a unified methodology [17]. Classification according to the SEUROP classification system uses a five-point scale (E–P) in the case of dual-purpose cattle. The highest meat yield is represented by class E and the lowest yield by class P. The grades are converted to numbers (E = 1 to P = 5) for the purpose of calculation in this analysis. A scale of 1 to 5 is used to assess the degree or level of fatness, where the lowest degree of fatness of the carcass is indicated by a 1 and the highest is indicated by a 5. No animal in our dataset obtained a value 5 level of fatness. Muscularity (MUS) is an exterior trait, subjectively assessed by the evaluator. Grade 1 means extremely concave haunch (poor muscularity) and conversely grade 9 means extremely convex haunch (excellent muscularity).

All the results for F and S tests were processed separately. Testing of individual factors affecting meat yield traits took place using the GLM method by SAS 9.1 statistical software [18].

### 2.2. Phenotypic Analysis


(a)Station test (S):
y_ijklmno_ = µ + b × age + hys_i_ + mus_j_ + fat_k_ + calv_l_ + btype_m_ + syear_n_ + smon_o_ + e_ijklmno_(1)(b)Field test (F):y_ijklmno_ = µ + b × age + hys_i_ + slt_j_ + fat_k_ + calv_l_ + btype_m_ + syear_n_ + smon_o_ + e_ijklmno_(2)
where y is a dependent variable (CW, BW, CDG, DP, EUR, MUS, FAT), µ is the population mean and b is the regression coefficient expressing the relationship between the variable y and age (age). The contemporary factor (hys) represents the herd or station where the bulls are fattened, the year of birth and the season of birth (seasons: January to March, April to June, July to August and October to December). In the case of station tests, the muscularity (mus) factor, which consists of 8 grades, also enters the model. Alternatively, in the case of field tests, the place of slaughter of the bull (slaughterhouse, slt), consisting of 14 grades, is included in the model. The degree of fatness (FAT) factor consists of 4 grades. The muscularity and degree of fatness factors were not utilized in the models, where fatness and muscularity represented dependent variables. The following factors were also included in the model: the order of the cow’s lactation at the time of the bull calf’s birth (calv, 10 grades), the factor of birth of the bull calf as a singleton or part of twins (btype, 2 grades), year of slaughter (syear, 15, or rather 6 grades in the field test) and the month of slaughter (smon, 12 grades). The factors and their frequency are given in table No. 2. A TYPE III Sum of Squares using the GLM procedure was used during analysis [18].


The factors that had a statistically significant effect on meat yield traits (*p* < 0.05) were then entered into model equations for the purpose of estimating genetic parameters and breeding values using the REML method by REMLF90 software package [19]. The variance components were estimated using a multi-trait animal model, applied separately for the S method (seven traits) and for the F method (four traits). The genetic correlations for traits represented in both sources of data (CDG, CW, EUR and FAT) were estimated through bivariate analyses, e.g., CDG was observed in both methods (S and F), and we used CDG (F) and CDG (S) in a two-trait REML model and estimated the genetic correlation between CDG (F) and CDG (S). The same procedure was used for CW, EUR and FAT.

### 2.3. Genetic Parameters Estimation

For genetic parameter estimation, the following model was adopted:y = Xb + Zu + e(3)
where y is the vector of observation of selected meat yield traits, b is the vector of environmental (non-genetic) fixed factors, u is the vector of the random additive genetic effect of an individual (BV), e is the vector of random residual effect and X and Z are incidental matrices connecting the fixed factors to observation. In the case of the additive genetic effect, it was assumed that u ~ N(0, G⊗A), where G is the variance-covariance matrix between the genetic effects of individuals for each of the traits (in the case of the station test a 7 × 7 matrix, and in the case of the field test a 4 × 4 matrix). Matrix A is the relationship matrix expressing kinship relations between all individuals in the pedigree; three generations of ancestors were available leading to a pedigree of 35,906 individuals in S and 33,823 individuals in F. Residuals were assumed as e ~ N(0, R⊗I), where R is the variance-covariance matrix of the residual effects of individual traits and I is the unit matrix corresponding in dimensions to the number of individuals with yield.

Heritability for individual traits was calculated based on the estimated variances as:H^2^ = σ_g_^2^/(σ_g_^2^ + σ_e_^2^)(4)
where σ_g_
^2^ is the additive genetic variance and σ_e_
^2^ is the residual variance. Genetic correlations (r_g_) were calculated as:r_g_ = (σ_g1;g2_)/(σ_g1_ × σ_g2_)(5)
where ***σ_g1;g2_*** is the genetic covariance between two selected traits and σ_g1_ a σ_g2_ represent the genetic standard deviation of trait 1 and trait 2, respectively.

## 3. Results and Discussion

### 3.1. General Characteristic of Beef Traits

Table 1 contains descriptive statistics of the monitored meat yield traits according to the F (Field) and S (Station) test methods. The S method includes seven meat traits and the field test F method includes four meat traits. The group of animals included in the analysis consisted of 5176 individuals (the MUS indicator was only rated at FCS stations from 2007) for the S method and 9378 animals for the F method. The values of monitored variables differed with various intensity in both groups. The average age of the bulls at slaughter was 89.9 days fewer in the case of the S method compared to the F method. Our established average age at slaughter values correspond to those established by [10,20,21]. The average CW values in our case correspond to [10,21,22]; however, these are significantly lower than [20] and, conversely, higher than [23]. Similar values were found in the case of average BW values [10], while higher values for the Czech Fleckvieh breed are registered in [7,8]; however, lower values have also been recorded [16]. Higher CDG or comparable values are demonstrated in [1,9,20] or [21]. The average DP value of all monitored animals (just the S method) was 54.16%. Our established DP value is very low; other authors obtained a slightly higher value [9,10] or a much higher value [23]. In relation to the classification of meat yield according to the SEUROP classification system, our results did not differ markedly from those given by [9,21], which range around class R (3), similar to our case. In relation to the FAT trait, [9,21] obtained slightly higher values than ours.

Of the traits that were included in the model as dependent variables, a greater difference in relation to the average values can be observed in CW, which is 63 kg higher in the F method. On the other hand, some variables differed only very slightly; for instance, CDG varied by just 16 g in the F method. A larger number of meat yield traits (MUS, DP and BW) was evaluated in the S method. The greatest variability in the S method was the variability coefficient in relation to FAT (19.16%) and the lowest in relation to DP (4.13%). In the F method, the lowest variability coefficient was found in relation to age at slaughter (7.66%), in the case of dependent variables in relation to CW (11.47%) and the greatest in relation to EUR (17.86%).

The obtained lower values of age at slaughter and CW in the S method, compared to the F method, are caused by the methodology for fattening bulls at FCS stations [3,9], where the animals from the stations are slaughtered at a lower age than animals slaughtered commercially at slaughterhouses. A lower meat yield according to the SEUROP classification system (EUR) with slightly higher FAT is observed in relation to the station method (S), which may also be caused by the aforementioned lower age at slaughter. The effect of age at slaughter on meat yield indicators is discussed in Section 3.2.

Table 2 gives the distribution of bulls according to individual factors used in the model. This includes the number of calves born annually from 2004 to 2017 for the S method and from 2012 to 2017 for the F method. The numbers of individual stations or farms (1–6) and individual field tests (FCS) (1–3) are also included. The cow’s lactation number upon the birth of the bull calf was divided into 10 categories according to the year of lactation, with the understanding that the last category includes the 10th and all subsequent lactations. The birth type of the bull calf (as a singleton or as one of twins) is indicated using the following coding: 1 indicates that the calf was born a singleton, and 2 indicates the calf was born as part of a set of twins.

In the case of the S method, a specific FCS appeared in the HYS factor as the place of fattening. The animals always went to a specific slaughterhouse which is associated with a specific FCS. This means that the factor of the site of slaughter (slaughterhouse) was not included in the linear modelling for the S method. In the case of six farms, the animals went to various (total 14) slaughterhouses as indicated in the F method.

Figure 1 shows the frequency of the rating for muscularity, which is only evaluated at FCS fattening stations and ranges from within 2 points (3 cases) to 9 points (661 cases). Data show that the greatest number of animals (1960) was rated grade 7, or above-average muscularity.

The distribution of the ratings of carcass quality as per the SEUROP classification of carcass quality are illustrated in Figure 2. The data show the comparison between the S and F methods. It is clear that the greatest discrepancy in the number of cases is in the U class (316 vs. 2692). There is also a higher frequency in the case of the R class in the F method, and conversely, a higher frequency of the O class in stations. The SEUROP classification is a subjective rating, where animals are rated by a greater number of assessors, particularly in the F method, which may affect actual evaluation. The lower age at slaughter may also result in lower meat yield values in the S method. The frequency of classification according to SEUROP obtains better results in the Fleckvieh breed than in dairy breeds, e.g., the Polish Holstein breed has the highest frequency of bulls in class O and no animal in class E [24,25]. On the other hand, the frequency of classification of first-class beef breeds (Piemontes) ranges in the opposite values according to SEUROP. Carcass analysis of Piemontes bulls in Italy gives classes according to SEUROP ranging from S+ to U [20].

Considering the greater size of the group of bulls fattened at FCS, it can be stated that the trends observed for individual classes for the degree of fatness (Figure 3) is comparable for both methods. Bulls in method F only show higher values in relation to the Grade 2 fatness. Nearly identical classification ratings for the degree of fatness are also given by other authors [25,26].

### 3.2. Effects of Environmental Factors

Table 3 presents the statistically conclusive effects of individual non-genetic factors on meat yield indicators. Various factors for estimating genetic parameters were chosen on the basis of this conclusiveness. Averages (LS MEANS) for individual levels of all factors for method S (Table 4) and for method F (Table 5) show a degree of variance and trends complying with statistical significance according to Table 3.

The following factors were included as variables:

Age at slaughter: the regression relationship between meat yield traits and age at slaughter in days was used in both methods (S, F) as a simple linear regression model (polynomial regression coefficients appeared in the models as statistically inconclusive and practically zero). The influence of age at slaughter was always statistically highly significant in relation to all meat yield traits. In the case of the F method we can see a decline in CDG along with the rising age of the bull, which falls from values of over 800 g at age 350 days to approximately 450 g at age 850 days (regression coefficient b = −0.6996). It must be noted that the age at slaughter, particularly in the S station method, had a very narrow range (see Table 1). This is why it is not possible to model a more precise curve shape for CDG or the gain in body weight in the S method, for example. In addition, BW (b = 1.1654) and CW (b = 0.1581 in the F method, alternatively 0.6548 in the S method) show a rising trend with growing age, as expected. Classification ratings according to the SEUROP classification (b = −0.0026 and −0.0011, respectively) and DP (b = 0.0040) were slightly better correlated with greater age. The FAT percentage (b = 0.0018 and 0.0004, respectively) increases with growing age and MUS (b = 0.0218) also improves slightly. The age at slaughter is given as a statistically significant factor by a number of authors, e.g., [14,15,20]. According to these authors, both the carcass weight and fatness increase with growing age [14]. A significant increase in weight and reduction in daily weight gain at the end of the fattening period was confirmed by [7,8].

HYS: This contemporary factor, incorporating herd (FCS) and year and season of birth, was statistically significant in relation to all the observed traits, as expected. This effect, similarly to the assessment of milk yield, explains a significant part of the variability since it includes the rearing environment, nutrition and seasonality. The conclusive influence of the season and year of birth on meat yield indicators is stated by [15]. The conclusive influence of the herd, year of birth and season on selected meat yield traits is also stated for various breeds [14,27].

Calving number: the mother’s lactation number during the birth of the calf appears a significant factor only in relation to CDG (in both methods), and also in relation to DP and CW (only for the F method). It can, therefore, be stated that the mother’s lactation number during the birth of the calf has an effect, particularly in the case of CDG. On the basis of LSMEANS from Table 4, there is no evident significant difference in the averages for all monitored meat yield indicators. The mother’s lactation number appeared significant in relation to body weight [27,28], but this factor conversely had an inconclusive statistical effect on CDG [28].

Birth type: the effect of whether the bull was born as a singleton or as part of a set of twins was statistically significant in relation to all traits in the F method. A statistically significant effect was not demonstrated in relation to DP and EUR in the S method. Findings suggest that whether the bull was born as a singleton or as part of a set of twins has a specific significance during growth and development and the subsequent yield from the animal during slaughter. Singletons had a tendency towards higher BW, CW, CDG and MUS values (Table 4). There is no statistically significant difference in the average values for FAT, EUR and DP. An analysis of meat yields in the Charolais breed [27,29] also demonstrated the significant effect of birth type on the animals’ weight indicators. The effect of birth type is included in the model for estimating the breeding values of meat breeds in the Czech Republic, for example [12].

Year and month of slaughter: the year and month of slaughter was not statistically significant except in relation to DP and EUR in the S method. In the case of the F method the year of slaughter was not statistically significant at all. On the contrary, the month of slaughter was statistically significant in relation to all traits. Some studies also include the effect of the year and month of slaughter as a statistically significant factor [21]

Slaughterhouse: the influence of the site of the slaughter house, only evaluated in the F method, was always statistically significant. As well as other authors, the effect of the site of slaughter was included in [14,21].

MUS: the muscularity effect was only evaluated for the S method and, with the exception of when it appeared as a dependent variable, it was always used as a fixed factor, which was always statistically significant. The average values from Table 4 simply document the positive relationship between MUS and BW, CW, CDG and DP (Table 4). On the contrary, in the case of classification according to the SEUROP classification, the relationship was negative and no evident trend was observed in relation to FAT.

FAT: similar to muscularity, the degree of fatness (FAT) was used as a fixed explanatory factor for meat yield indicators and was also always statistically significant. A higher BW, CW and CDG was evident with a higher FAT value. While no evident trend was observed in relation to DP and EUR.

The averages in Table 5 (method F) show the same trend as the averages from Table 4 (S method) in relation to all the factors included in the model.

### 3.3. Estimation of Genetic Parameters

It is evident from Table 6 that the mean values of the heritability in relation to DP, CW and CDG were calculated for the S station method. Slightly lower h^2^ values (0.27, respectively 0.26) were estimated for BW, or rather MUS. The lowest values were registered for FAT (0.20) and EUR (0.18). The newly established heritability for FAT, ranging between 0.17–0.20, are evidence of their potential application in breeding programs selecting for meat yield.

Our results agree with the findings of other authors on these aspects. Heritability (h^2^) values of 0.14–0.26 for body weight and 0.34 for daily gain were established for Norwegian Red Bulls [13], while h^2^ values of 0.20 for FAT; 0.28 for CW [30] were given for the Hanwoo breed in Korea. The h^2^ value for EUR and CDG was estimated identically as 0.25 [1] for the Piemontese intensive meat breed in Italy. In another study of the Piemontese breed population in Italy, the authors reported an h^2^ value for EUR of 0.07, a value of 0.32 for CDG and 0.19 for CW [20]. In the case of Canadian Charolais, the h^2^ value for CW was 0.32 and 0.38 for FAT [22]. An h^2^ value of 0.16 [21] for CDG was estimated for the related Fleckvieh population in Germany. Variable heritability estimates were also reported for weight gain ranging from 0.16 (station test) to 0.22 (field test) for beef breeds in Sweden [2]. These values were estimated under the different conditions associated with field and station tests. A heritability for FAT of 0.35 and 0.50 for CW is described for Korean Brindle Cattle [14]. In the case of cattle reared in the Czech Republic, h^2^ values range from 0.20 to 0.35 for body weight (BW) [15]. Moderately high h^2^ for CW and moderately low for DP are reported by [31].

The interrelationships between heritability values, genetic correlations and breeding values calculated for the S method are reported in Table 6.

Genetic correlations (r_g_) show strong genetic interrelationships between CDG, CW, BW and MUS (r_g_ is greater than 0.9). The SEUROP classification demonstrates strong negative correlations towards all monitored parameters, except for FAT (from −0.60 to −0.76). It must be noted that the numerical expression of FAT traits means that a lower numerical value is linked to a higher SEUROP class for meat yield. This basically means that this is a positive statistical relationship between the aforementioned traits and the SEUROP classifications. On the contrary, meat yield, as described in the SEUROP classification, does not demonstrate a more significant statistical relationship with regard to fatness (FAT), when compared to any of the other evaluated meat yield traits. The maximum r_g_ value (−0.29) is between fatness (FAT) and DP. The strong correlation between the ratings for exterior muscularity and important meat yield indicators (CDG, CW, BW and EUR), ranging from 0.79 to 0.97, is evidence of the importance of the evaluation of the related exterior characteristics of animals when breeding for meat yield.

In the case of correlation between estimated breeding values, one can observe the same trend as for r_g_, although with a slightly more marked dependence between individual meat yield traits, in the case of both positive and negative correlation coefficients.

The interrelationships between heritability values, genetic correlations and breeding values calculated for the F method are reported in Table 7.

An analysis of the interrelationships reported in Table 7 offers similar trends to those described in Table 6. There are a few differences, however. In the case of the heritability (h^2^), generally lower values were reported for all the observed traits. This may be because of better conditions for the test at the FCS (under the same conditions and strict methodology). The trend is also similar to that reported for method S with regard to r_g_ and r_bv_. When compared to method F, correlations reported in the station method demonstrate slightly higher values.

In compliance with our results, a very strong genetic correlation (r_g_ = 0.83) between BW and CDG [13] was also calculated. The genetic correlation between CW and FAT (r_g_ = 0.18) is stated by the authors [22]. Heritability (h^2^) was rated 0.6 [14] for the same traits in Korean Brindle cattle. An r_g_ value of 0.31 was calculated between FAT and CW [30] in another population of cattle in Korea (Hanwoo). An analysis of the Piemontese population in Italy established an r_g_ value of 0.88 between CW and CDG, an r_g_ value of −0.43 between CW and EUR and an r_g_ value of −0.65 between CDG and EUR [20].

Estimates of genetic parameters for individual pairs of traits, which occur in both the F method and the S method (Table 8), were calculated in order to establish possible similarities between the station and field test. These pairs (calculated for both methods F and S) were CDG, CW, EUR and FAT and appeared in the calculation as independent traits, which means that the genetic correlation (r_g_) between them was estimated. Heritability (h^2^), which confirm the higher values for method S, as mentioned earlier in Table 6 and Table 7, are also presented. A comparative genetic correlation (r_g_) value for CDG between the field and station tests of 0.75 [1], was established in Piemontese cattle in Italy. Groenveld et al. [32] pursued the link between the station and field tests for individual meat yield indicators in Slovakian pig population. They state a genetic correlation between the weight gain established by station and field tests in a value ranging from 0.42–0.53, i.e., a significantly lower value when compared to the values we established for cattle.

Calculations made for this study established a high genetic correlation between the individual S and F methods in the case of CDG and CW. This correlation is slightly lower for EUR and FAT, which indicates that the methods are only interchangeable for CDG and CW. There is a specific difference evident between the individual methods in the case of EUR and FAT. This can partially be explained by the lower heritability, or the greater environmental effect on EUR and FAT. Additionally, the low r_g_ in EUR and FAT between the F and S method may be due to the fact that these traits, unlike CDG and CW, are evaluated subjectively. CDG and CW are based on a weight measurement that is exact (weighing at the slaughterhouse—F—or at the station—S), whereas EUR and FAT are rated by evaluators (in the case of the F method by a large number of evaluators in different slaughterhouses, in the case of the S method by a limited number of professionally trained staff in FCS stations). The lower r_g_ between EUR of F and EUR of S and between FAT of F and FAT of S suggests that these indicators need to be considered as different traits in genetic evaluation. Thus, the selection for FAT in FCS (S method) may not be the same as the selection based on FAT in slaughterhouses (method F). The same could be applied for EUR. Harmonization of evaluators is important, otherwise it would be necessary to maintain a multivariate genetic analysis taking into account the data source (FCS or field test).

### 3.4. Breeding Values and Genetic Trends of Beef Traits

According to theoretical assumptions, the breeding value (BV) average ranges around zero and obtains the values of standard deviations corresponding to the variability of the relevant traits (see Table 1). The EUR, FAT, MUS and DP indicators obtain the lowest variability, or rather determinate deviation. The same trend is evident in both methods, S and F. Přibyl et al. [12] evaluated averages and SD breeding values for beef yield in the Czech Republic. The average BV obtained various values of SD, for various breeds (multibreed multi-trait animal model), which corresponded to the values obtained in our work. The breeding values for the Fleckvieh breed were analyzed in the German population and our established breeding values demonstrate a much greater range than the German population [21].

Genetic gain can be expressed as the average breeding value according to the individual years of birth of the animals. Sires of the slaughtered bulls, who were or are used for insemination, and therefore, have a greater influence on breeding the population of dual-purpose cattle in the Czech Republic, were chosen from the pedigree for describe genetic gain. The average values of their breeding values according to year of birth are given in Figure 4 (method S) and Figure 5 (method F). We converted breeding values to relative breeding values with the standardized mean 100 and standard deviation 12, for comparing the genetic gain in different traits. Transformation on relative BV (RBV) was conducted using this formula:RBV = ((BV − mean_BV_)/SD_BV_ × 12) + 100(6)

A positive trend is apparent in traits related to quantitative expression of meat yield, particularly those related to body size (CDG, CW and BW) in the case of both methods. A positive trend is also apparent in MUS and DP. There is negative genetic gain in relation to the EUR. This can be explained by a lower heritability in the case of EUR, but it can also be partially considered the result of insufficient selection intensity on EUR and also in relation to the breeding goal of dual-purpose breeds, when genetic progress in relation to CDG, BW and CW is indirectly increased through increasing body size (and thereby also body weight) thanks to selection for dairy yield. CDG is one of the most important meat yield traits, which is used for selection in the population of Czech Fleckvieh cattle.

## 4. Conclusions

An analysis of 9378 bulls from the field test and 6346 bulls using the station method shows that environmental, non-genetic factors appeared to have the most significant effect on the monitored meat yield traits. This corresponds to the heritability values, which ranged from 0.14 to 0.33 of the total variability. If we compare the station method to the field test (F), we can state that the station method (S) generally provides higher genetic parameter values, but we can assume increased costs and increased labor compared to the field test (F). The strong positive relationship between the linear description for muscularity and significant meat yield traits (CDG, CW, BW and EUR) was also confirmed, which is evidence of the importance of an evaluation of the exterior (related characteristics) of animals when breeding for beef yield. The newly established heritability for FAT, ranging between 0.17–0.20, is evidence of its potential application in selection for meat yield.

The results of our study suggest that positive genetic trends are only found in meat yield traits that are related to the size or possibly the weight of the animal. These findings suggest important potential principles and criteria that can be applied and included in breeding programs when breeding dual-purpose cattle breeds. Potentially, we can genetically improve Fleckvieh in both main traits (meat and milk) simultaneously through body size and body weight, although it is generally assumed (among breeding professionals) that there is a negative interrelationship between milk and meat traits.

## Figures and Tables

**Figure 1 animals-10-02159-f001:**
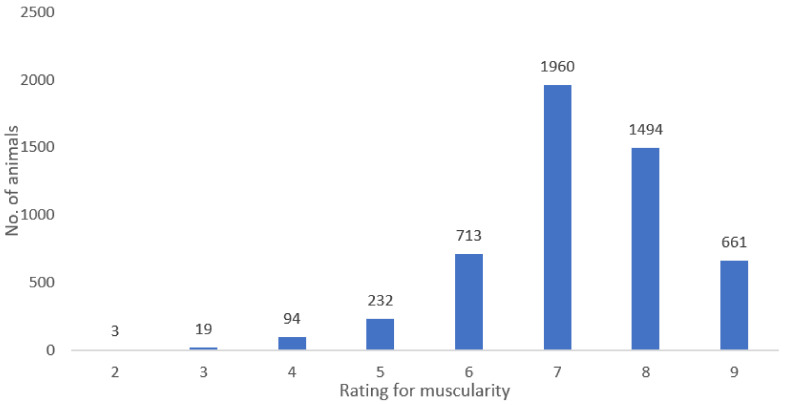
Frequency of the muscularity rating (MUS) (only in the S method).

**Figure 2 animals-10-02159-f002:**
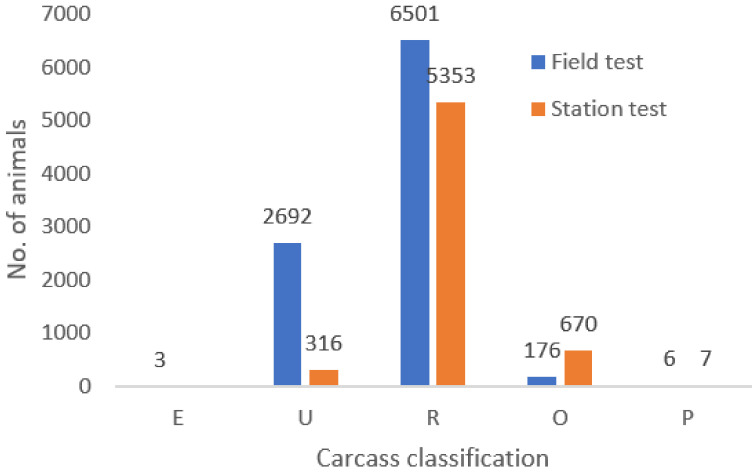
Frequency distribution—Carcass classification.

**Figure 3 animals-10-02159-f003:**
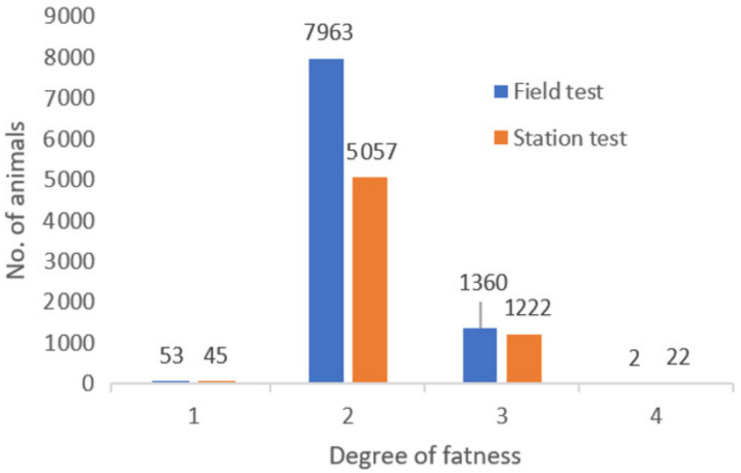
Frequency distribution—Degree of fatness.

**Figure 4 animals-10-02159-f004:**
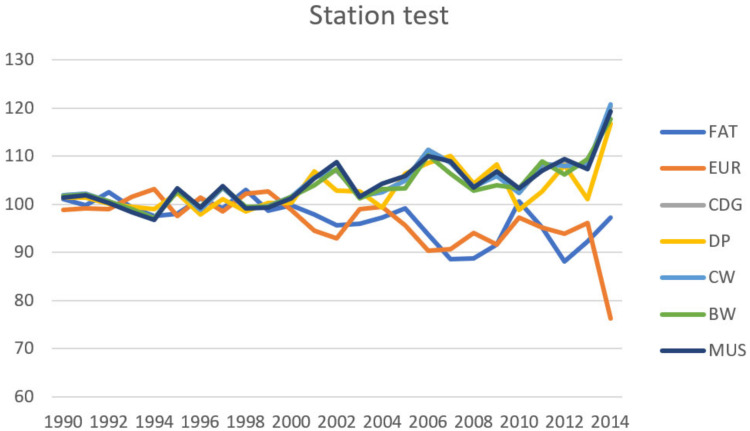
Genetic gain of the fathers of bulls (1208 bulls)—station test (S). FAT—degree of fatness, EUR—SEUROP classification, CDG—carcass daily gain, DP—dressing percentage, CW—carcass weight, BW—body weight, MUS—muscularity.

**Figure 5 animals-10-02159-f005:**
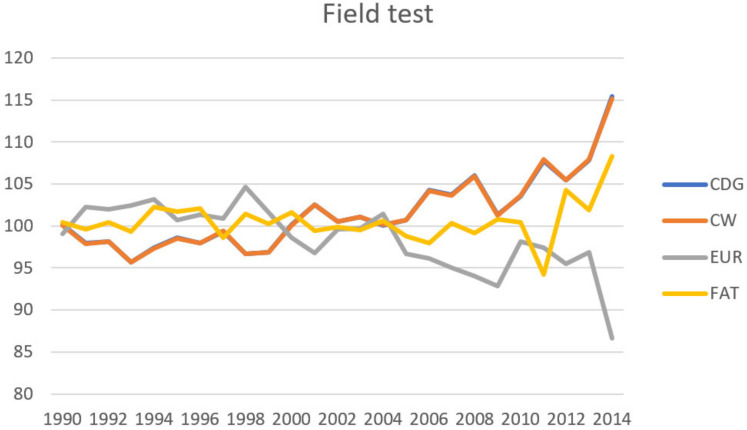
Genetic gain of the sires of bulls (1144 bulls)—field test (F). FAT—degree of fatness, EUR—SEUROP classification, CDG—carcass daily gain, CW—carcass weight.

**Table 1 animals-10-02159-t001:** Descriptive statistics of independent variables included in the two primary data sets.

Items	Units	No.	Mean	Min	Max	SD	S.E.	CV
Station Test (S)								
Age at slaughter	days	6346	556.39	513.00	620.00	37.64	0.47	6.77
Degree of fatness	point	6346	2.19	1.00	4.00	0.42	0.01	19.16
Muscularity	point	5176	7.25	2.00	9.00	1.14	0.02	15.69
SEUROP	point	6346	3.06	2.00	5.00	0.40	0.00	12.93
Carcass daily gain	g/day	6346	609.74	335.00	849.00	71.13	0.89	11.67
Dressing percentage	%	6346	54.16	39.70	70.20	2.24	0.03	4.13
Carcass weight	kg	6346	339.37	176.88	497.92	46.32	0.58	13.65
Body weight	kg	6346	625.97	359.51	911.24	77.25	0.97	12.34
Field Test (F)								
Age at slaughter	days	9378	646.32	385.00	795.00	49.51	0.51	7.66
Degree of fatness	point	9378	2.14	1.00	4.00	0.36	0.00	16.98
SEUROP	point	9378	2.73	1.00	5.00	0.49	0.01	17.86
Carcass daily gain	g/day	9378	626.05	343.00	934.00	77.95	0.80	12.45
Carcass weight	kg	9378	402.91	250.10	545.89	46.20	0.48	11.47

CV—coefficient of variability (%), S.E.—standard error, SD—standard deviation, Min—minimum value/count, Max—maximum value/count.

**Table 2 animals-10-02159-t002:** Frequency distribution of bulls according to individual factors as applied in the model.

Items	Station Test (S)	Field Test (F)
	No.	No.
Birth year		
2004	659	
2005	757	
2006	694	
2007	541	
2008	381	
2009	471	
2010	479	
2011	430	
2012	364	1523
2013	406	1693
2014	372	1630
2015	356	1477
2016	259	1448
2017	177	1607
Station/Field		
1	800	1983
2	3769	1002
3	1777	795
4		1544
5		2414
6		1640
Calving number		
1	1011	2692
2	2651	2154
3	1329	1726
4	615	1293
5	364	744
6	191	413
7	105	200
8	43	100
9	21	39
10	16	17
Birth type		
1	6138	8911
2	208	467

Birth type (1 = normal calving; 2 = twins).

**Table 3 animals-10-02159-t003:** Effect of non-genetic factors on beef traits (analysis of covariance—MSS).

Effects	DF	Station Test (S)							DF	Field Test (F)			
		Carcass Daily Gain	Carcass Weight	Body Weight	Dressing Percentage	SEUROP	Muscularity	Degree of Fatness		Carcass Daily Gain	Carcass Weight	SEUROP	Degree of Fatness
Age at slaughter	1	49,460.25 **	6278.38 **	21,068.69 **	142.12 **	0.37 *	7.93 **	0.44 *	1	1,233,704.86 **	195,985.39 **	7.36 **	0.55 *
Herd-year-season	133	23,154.41 **	6565.75 **	16,682.95 **	45.74 **	0.59 **	22.58 **	0.68 **	143	6,789,928.07 **	19,164.46 **	1.21 **	0.71 **
Calving number	9	3813.29 *	1026.79 ns	2951.83 ns	6.18 *	0.072 ns	1.07 ns	0.14 ns	9	81,250.01 **	3783.41 **	0.26 ns	0.08 ns
Birth type	1	8133.77 *	3031.51 *	5374.62 *	6.83 ns	0.15 ns	10.10 **	0.35 ns	1	129,355.58 **	59,703.13 **	1.44 **	0.47 *
Year of slaughtering	14	5147.26 **	1579.25 **	4820.98 **	2.04 ns	0.08 ns	19.87 **	0.29 *	6	23,339.10 ns	1472.47 ns	0.33 ns	0.19 ns
Month of slaughtering	11	3311.01 *	1026.56 *	4141.09 **	2.25 ns	0.08 ns	18.53 **	0.32 *	11	105,501.13 **	3657.91 **	0.42 *	0.77 **
Slaughterhouse	-	-	-	-	-	-	-	-	13	3,401,478.91 **	97,754.09 **	10.16 **	0.98 **
Muscularity	8	1,399,133.06 **	437,778.05 **	1,208,601.22 **	130.84 **	33.71 **	-	1.75 **	-	-	-	-	-
Degree of fatness	3	70,689.65 **	22,386.81 **	112,909.48 **	25.05 **	1.77 **	24.73 **	-	3	494,179.07 **	215,124.88 **	2.48 **	-
Error	6 165	1,862.30	565.68	1804.21	2.85	0.09	0.94	0.15	9 190	2982.55	1225.92	0.19	0.12
RMSE		43.15	23.78	42.48	1.69	0.30	0.97	0.39		54.61	35.01	0.44	0.34
R-Square		0.64	0.74	0.71	0.45	0.45	0.87	0.17		0.52	0.44	0.20	0.13

* *p* < 0.05; ** *p* < 0.01; ns, not significant; MSS, mean sum of square, DF, degree of freedom; RMSE, root mean square error; R-square, % of explained variability.

**Table 4 animals-10-02159-t004:** Least square means and standard error of traits by selected environmental factors—station test (S).

Items	Level	Body Weight		Carcass Weight		Carcass Daily Gain		Dressing Percentage		SEUROP		Degree of Fatness		Muscularity	
		LS Means	s.e.	LS Means	s.e.	LS Means	s.e.	LS Means	s.e.	LS Means	s.e.	LS Means	s.e.	LS Means	s.e.
Muscularity	2	423.07	26.11	224.09	14.62	394.73	26.53	52.01	1.04	4.43	0.18	2.15	0.24		
	3	453.98	12.64	237.96	7.08	423.30	12.84	52.02	0.50	4.24	0.09	2.03	0.11		
	4	489.53	9.16	256.83	5.13	463.94	9.30	52.41	0.36	4.12	0.06	1.94	0.08		
	5	526.05	8.52	278.76	4.77	501.45	8.66	52.97	0.34	3.88	0.06	2.04	0.07		
	6	556.74	8.22	297.33	4.60	535.35	8.35	53.43	0.33	3.48	0.06	2.13	0.07		
	7	606.67	8.11	326.95	4.54	587.62	8.24	53.95	0.32	3.25	0.06	2.18	0.07		
	8	660.05	8.11	358.91	4.54	644.97	8.24	54.40	0.32	3.02	0.06	2.23	0.07		
	9	712.70	8.31	392.18	4.65	704.72	8.45	54.95	0.33	2.86	0.06	2.24	0.07		
Degree of fatness	1	542.71	10.30	290.65	5.77	524.42	10.46	53.29	0.41	3.76	0.07			5.85	0.23
	2	552.75	8.17	298.37	4.57	534.91	8.30	53.62	0.32	3.54	0.06			6.70	0.17
	3	572.39	8.25	306.97	4.62	550.41	8.38	53.33	0.33	3.48	0.06			6.92	0.17
	4	586.53	12.28	313.29	6.88	559.79	12.48	53.15	0.49	3.62	0.09			7.07	0.27
Calving number	1	562.85	8.52	301.68	4.77	541.17	8.66	53.32	0.34	3.61	0.06	2.12	0.07	6.65	0.18
	2	565.23	8.44	302.42	4.73	542.25	8.58	53.21	0.34	3.60	0.06	2.15	0.07	6.70	0.18
	3	565.76	8.52	302.98	4.77	543.52	8.65	53.26	0.34	3.61	0.06	2.15	0.07	6.68	0.18
	4	567.67	8.58	303.37	4.80	544.29	8.72	53.16	0.34	3.59	0.06	2.12	0.07	6.66	0.18
	5	567.43	8.70	304.36	4.87	546.28	8.84	53.34	0.35	3.62	0.06	2.13	0.08	6.59	0.19
	6	566.08	9.04	301.18	5.06	540.09	9.18	52.95	0.36	3.56	0.06	2.15	0.08	6.62	0.19
	7	556.81	9.40	298.73	5.26	535.50	9.55	53.33	0.37	3.60	0.07	2.13	0.08	6.71	0.20
	8	569.29	10.61	307.55	5.94	553.25	10.78	53.76	0.42	3.56	0.07	2.15	0.09	6.42	0.23
	9	571.05	12.62	310.84	7.06	558.24	12.82	54.03	0.50	3.62	0.09	2.03	0.11	6.82	0.28
	10	543.78	13.69	290.10	7.67	519.22	13.91	53.11	0.54	3.62	0.10	2.12	0.12	6.47	0.31
Birth type	1	566.24	8.43	304.31	4.72	545.64	8.56	53.44	0.33	3.58	0.06	2.10	0.07	6.75	0.18
	2	560.95	8.99	300.33	5.04	539.13	9.14	53.25	0.36	3.61	0.06	2.15	0.08	6.52	0.19

LS means—least square means, s.e.—standard error of the mean.

**Table 5 animals-10-02159-t005:** Least square means and standard error of traits by selected environmental factors—field test (F).

Items	Level	Carcass Weight		Carcass Daily Gain		SEUROP		Degree of Fatness	
		LS Means	s.e.	LS Means	s.e.	LS Means	s.e.	LS Means	s.e.
Degree of fatness	1	341.43	6.82	533.52	10.64	2.95	0.09		
	2	369.70	4.08	573.38	6.36	2.86	0.05		
	3	382.97	4.20	593.74	6.54	2.78	0.05		
	4	384.76	25.37	595.33	39.57	2.68	0.32		
Calving number	1	368.13	4.01	570.96	6.25	2.87	0.05	2.06	0.04
	2	370.80	4.01	575.10	6.25	2.89	0.05	2.08	0.04
	3	371.96	4.02	576.84	6.26	2.88	0.05	2.07	0.04
	4	372.94	4.04	578.40	6.30	2.87	0.05	2.08	0.04
	5	372.06	4.16	576.95	6.48	2.87	0.05	2.07	0.04
	6	370.49	4.32	574.83	6.73	2.92	0.05	2.07	0.04
	7	364.93	4.68	566.21	7.29	2.91	0.06	2.10	0.05
	8	367.04	5.33	569.34	8.31	2.89	0.07	2.07	0.05
	9	367.95	6.94	569.58	10.82	2.79	0.09	2.10	0.07
	10	376.88	9.55	585.74	14.89	2.70	0.12	1.98	0.09
Birth type	1	375.99	4.06	582.74	6.33	2.83	0.05	2.05	0.04
	2	364.64	4.34	566.05	6.76	2.89	0.05	2.09	0.04

LS means—least square means, s.e.—standard error of the mean.

**Table 6 animals-10-02159-t006:** A matrix relating the heritability values (h^2^—diagonal), genetic correlations (r_g_—above the diagonal) and correlations of breeding values (b_v_—below the diagonal) reported for the Station (S) method.

Trait	FAT	EUR	CDG	DP	CW	BW	MUS
**FAT**	0.2024	0.0032	0.0002	−0.2885	0.0058	0.1133	−0.0229
**EUR**	0.1572	0.1761	−0.7495	−0.6023	−0.7681	−0.6414	−0.6997
**CDG**	−0.1640	−0.8408	0.3023	0.5082	0.9972	0.9386	0.9239
**DP**	−0.4141	−0.7220	0.6104	0.3341	0.4983	0.1880	0.3805
**CW**	−0.1586	−0.8554	0.9976	0.5961	0.3133	0.9447	0.9170
**BW**	−0.0317	−0.7263	0.9418	0.3152	0.9496	0.2742	0.8945
**MUS**	−0.1413	−0.7850	0.9665	0.5020	0.9597	0.9378	0.2672

FAT—degree of fatness, EUR—SEUROP, CDG—carcass daily gain, DP—Dressing percentage, CW—Carcass weight, BW—Body weight, MUS—Muscularity.

**Table 7 animals-10-02159-t007:** A matrix relating the heritabilities (h^2^—diagonal), genetic correlations (r_g_—above the diagonal) and correlations of breeding values (b_v_—below the diagonal) reported for the Field (F) method.

Trait	CDG	CW	EUR	FAT
**CDG**	0.2656	0.9987	−0.5972	−0.0261
**CW**	0.9995	0.2557	−0.5945	−0.0380
**EUR**	−0.7607	−0.7568	0.1430	0.0617
**FAT**	−0.0460	−0.0584	0.0705	0.1673

FAT—degree of fatness, EUR—SEUROP, CDG—carcass daily gain, CW—Carcass weight.

**Table 8 animals-10-02159-t008:** Comparative genetic correlation (r_g_) and heritability (h_2_) values calculated for beef traits from the station (S) and field (F) test.

Testing Method		CDG	CW	EUR	FAT
**Field test/station test**	r_g_	0.8351	0.8244	0.5818	0.5676
**Field test**	h^2^	0.2835	0.2794	0.1374	0.1595
**Station test**	h^2^	0.3013	0.3031	0.1610	0.2103

FAT—degree of fatness, EUR—SEUROP, CDG—carcass daily gain, CW—Carcass weight.

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
