# Peer review of "Environmental Factors and Genetic Parameters of Beef Traits in Fleckvieh Cattle Using Field and Station Testing"

_animals, 2020, doi:10.3390/ani10112159_

Round 1

Reviewer 1 Report

I have no further comments.

Author Response

Thank you very much for review. 

Reviewer 2 Report

Please see the PDF attached

Author Response

1 Question/comment:

 I feel a closing sentence (a sort of conclusion) is missing in the abstract (line 37).

We added sentences in lines 38-42.

2 Question/comment:

Also overall, try to be consistent with the use of the F for field testing and the S for station testing (e.g. line 23, lines 30-31, line 70, line 109, lines 114-115, line 220, etc.) otherwise the reader may get confused.

We uniformed using of the F and S through the manuscript.

3 Question/comment:

The first part of the introduction (lines 41-49) desribes briefly aims and methodology of the study but usually the details are placed at the end of the introduction.

Part from lines 41-49 was moved to the end of manuscript (now lines 76-84).

4 Question/comment:

I would split up Materials and Methods (chapter 2) as: 2.1 Dataset; 2.2 Phenotypic Analysis; 2.3 Genetic parameters estiamtion

Chapter 2 was divided to the sections as you recommend (lines 86, 130, 182).

  1. Question/comment:

I have found the description of statistical analysis a bit mesed up. I would move up the section from line 122 to line 141, immediatelly after line 111. I did not understand why FAT was included as fixed effects in the phenotypic analysis of CW, BW, CDG, DP, EUR nad MUS. Please clarify. The same MUS for the variables FAT, CW, BW.

We moved up sentences according to the reviewer's comment (now lines 130 - 149).

FAT and MUS are used in phenotypic analysis because they had a significant effect on the other meat indicators (table 6) and also increased the R-square of the overall model (Table 6). However, these effects (FAT and MUS) could not be used when analyzed as depend varibales.

6 Question/comment:

 Info in lines 170 – 171 may be moved to the section 2.1 Dataset.

We would like let the sentence in current lines, because these numbers described tables 1 in chapter results and in chapter 2 (materials nad methods) was also included.

7 Question/comment:

In the tables try to avoid extended use of abbreviations; tables should somehow be self-explanatory. I recommend to use footnotes if necessary, e.g. to define birth type (1 = normal calving; 2 = twins).

Tables were corrected, we replace abbreviations where it was possible, else we used footnotes. See tables in track changes version of manuscript.

8 Question/comment:

I was concerned when reading table 2 and results in table 8: why not merging parity 5,6,7,8,9 and 10 in one class?

The frequency of observations at 5th – 10th lactation is low, but even in the smallest category there are 16 observations. For example, when estimating the BV of milk yield using the lactation animal model, was required at least 3-5 observations per 1 level of HYS effect in the analyses of the HYS effect (some farms of dairy cows are small and merging of farms could be problematic). However, I agree with the opponent that merging smaller groups of observation would be a better way.

9 Question/comment:

Please try to interpret and discuss a bit more the moderate genetic correlations found between the same trait when recorded in F and in S: the rg between EUR of F and EUR of S and between FAT of F and FAT of S (Table 5). Which practical recommendations would you make based on such results?

We add this paragraph into manuscript:

Also, the low rg in EUR and FAT between F and S method may be due to the fact that these traits, unlike CDG and CW, are evaluated subjectively. CDG and CW are based on a weight measurement that is exact (weighing at the slaughterhouse - F or at the station - S). Whereas EUR and FAT are rated by evaluators (in the case of the F method by a large number of evaluators in different slaughterhouses, in the case of the S method by a limited number of professionally trained staff in FCS stations). The lower rg between EUR of F and EUR of S and between FAT of F and FAT of S suggests that these indicators need to be considered as different traits in genetic evaluation. Thus, selection for FAT in FCS (S method) may not be the same as selection based on FAT in slaughterhouses (method F). The same could be applied for EUR. Harmonization of evaluators is important, otherwise it would be necessary to maintain a multivariate genetic analysis taking into account the data source (FCS or field test).

Additional comments of reviewer:

Lines 22-23: we have modified the sentence according to your recommendation.

Lines 26-27: we have modified the sentence according to your recommendation.

Line 32: we have modified the sentence according to your recommendation.

Line 50: we have modified the sentence according to your recommendation.

Line 51: we have modified the sentence according to your recommendation.

Line 56: we have modified the sentence according to your recommendation.

Line 78: we have modified the sentence according to your recommendation.

Line 87: we have modified the sentence according to your recommendation.

Line 89: we have modified the sentence according to your recommendation.

Line 92: we have modified the sentence according to your recommendation.

Line 95: we have modified the sentence according to your recommendation.

Line 103: Yes it was a typo, we drop it.

Line 104-105: We have the sentence in the same form you suggested.

Lines 113-114: we have modified the sentence according to your recommendation.

Line 116: we have modified the sentence according to your recommendation.

Line 119-121: we have modified the sentence according to your recommendation.

Line 142: we have modified the sentence according to your recommendation.

Line 147-149: we have modified the sentence according to your recommendation.

Line 149: we have modified the sentence according to your recommendation.

Lines 152-154: we have modified the sentence according to your recommendation.

Line 154: we have modified the sentence according to your recommendation.

Line 156-157: we have modified the sentence according to your recommendation.

Line 158: we have modified the sentence according to your recommendation.

Line 161: we have modified the sentence according to your recommendation.

Line 164: we have modified the sentence according to your recommendation.

Line 168: we have modified the sentence according to your recommendation.

Line 182-183: we have modified the sentence according to your recommendation.

This manuscript is a resubmission of an earlier submission. The following is a list of the peer review reports and author responses from that submission.

Round 1

Reviewer 1 Report

please check the attached file

Author Response

We answered the following questions/comments:

Comment/question:

It is hard to figure out to understand how carcass(net) daily gain observed.

Answer:

CDG is ratio of weight of carcass (def. of carcass, beef quarters: the muscle, bone and fat associated with the slaughter of an animal, left after the removal of the head, hide, legs and internal organs) to age of animal at slaughter. CDG = (weight of carcass in kg *1000) / age in days.

Expalnation was added in revised manuscript (lines 95-98).

Comment/question:

Page 2. 23-24 lines: it is also hard to understand what below sentence means.

carcass weight 402.91 kg, resp. 339.37 kg; carcass daily gain 626.05 g/day, resp. 609.74 g/day; SEUROP carcass classification 2.73, resp. 2.19.

Answer:

Sentence was corrected (see revised manuscript lines 22-25)

Comment/question:

  1. Degree of meat yield (EUR)–low meat yield (5) to high meatyield (1)
  2. Degree of fatness (FAT)-low fatness (1) to high fatness (5)
  3. muscularity rating (MUS)-?

Above sources of information were used as factors and/or traitsin the considered models.

What does means on the considered model?

Answer:

Muscularity – linear exterior trait, clasiffied by human examinator (1 – weak muscling… 9 excellent muscling of body). We added the explanation of scale of muscuarity into the manuscript (lines 105-108).

Degree of meat yield (EUR) was used only as depend variable (trait).

Muscularity and fatness were used as both, factors and depend variables (traits) only in GLM modelling. If fatness or muscularity was used in model as depend variable, these sources of information do not use as factors. We excluded fatness and muscularity as factors due to solving possibilities of MME (singularity of left-hand side matrix in MME) in case of multi-trait REML analysis.

Comment/question:

Page 5 line 108-110: it is also hard to understand what below sentence means.

The genetic correlations for traits represented in both sources of data (CDG, CW, EUR and FAT) were then estimated by always entering each trait in the model twice (depending on the source of data) and this being considered an individual trait.

Answer:

Sentence was corrected in manuscript (lines 117-120). The idea was as follows: traits CDG, CW, EUR and FAT were observed in both sources of information (slaghterhouses and FCS stations). We estimated genetic parameters for CDG using 2-trait model (first trait was CDG from slaughterhouse and second trait was CDG from FCS stations). So we established genetic correlations between these both CDG traits from different sources (FCS and slaghterhouses). This design we used aso for CW, EUR and FAT.

Comment/question:

Classifying level of effects was confused according models. Please make sure.

Answer:

We checked all effects in model and we found no mistakes. Can reviewer pointed to specific confusing clasyfying levels of effects?

Comment/question:

For estimating genetic (co)variances, FAT and MUS (Station data only) research work on this paper would be assumed as dependent variables on 7 and/or 4 multi traits model.

However, there was no explanation of genetic model. Please make clearly explain.

Answer:

The 7-trait model was used for station method, and 4 traits model for field (slaughterhouses system). The trait FAT was observed in both systems, so we estimated genetic parameters for FAT in 4 and 7-trait model. Trait MUS was observed only in station, so this trait was estimated only in 7-trait REML model (see corrected raws 105 – 110). Genetic model was the same for for all traits (chapter 2.2) and the fixed effect were chosen according to GLM testing – included effects p > 0.05 (table 6).

Comment/question:

EUR, MUS and FAT measurements were ordered categorical data, So I would like to recommend that you can do analysis using threshold model.

Answer:

We agree with reviewer that the variables EUR, MUS and FAT are categorical data and more accurate should be threshhold models. But as we know, routine estimating of breeding values in Czech republic for beef traits was is performed using linear animal model, whe SEUROP are translate to numeric data. We do the same in our manuscript. Comparing of using linear animal model and threshold animal model for cathegorical beef traits and exterior traits was desrcibed e.g. by Veselá et al., 2011 (DOI: https://doi.org/10.17221/1292-CJAS). Estimated heritabilities are comparable with our results.

Comment/question:

Heritability estimates on the tables represented genetic parameters were shown as percentage unit and genetic correlations were shown as unit. Please show unified format.

Answer:

We uniformed coefficients of heritabilities and genetic parameters (see revised manuscript, lines 27, 28, 348-351(tab 3), 370-372 (tab 4), 386-378 (tab 5) and line 458).

Reviewer 2 Report

Usually, we call correlation or correlation coefficient but do not call heritability coefficient. We just call heritability or heritability estimate(s).

Basically, CDG and CW are genetically the same trait. One of those traits can be removed. Other highly correlated traits (BW and MUS) can be also removed to reduce the evaluation cost if necessary.

Details:

Lines 23: What is “resp.”? Please spell out and then define it.

Lines 26-36: Heritability and correlations should be consistent by using the same number of decimals. 13.74% (=0.1374) and 33.41% (=0.3341) for heritability can be 14% (or 0.14) and 33% (or 0.33); 0.8351 and 0.8244 for correlation can be 0.84 and 0.82. Later, on the other hand, 0.17 – 0.20 and 0.79 – 0.97 were used.

Line 32: Does this “A genetic heritability evaluation” mean “heritability estimation” or “genetic evaluation”?

Lines 89-90: What are meat yield (DP) and net gain (CDG)? Please define them.

Line 98: Is the grade (E to P) a subjective measurement by human or an objective by a measure?

Line 155: “σg1 a σg2” are not variances.

Lines 439-442: If meat traits are negatively correlated with milk, how can both traits be genetically improved? Discussion is needed before conclusion, by showing some evidence or citing other studies for the same breed.

I suggest not to use percentage in heritability as used in Tables 3, 4, and 5 and some sentences in the text. In results and discussion, most of the results were not expressed in percentage. Also, the number of decimals for heritability estimates and correlations should be consistent, using always 2 decimals (0.xx).

In Table 9, we cannot compare SD for S and F because those values depend on estimates of variance components for additive genetic and residual effects. Also, we cannot compare the means between S and F because the range of years is different. So, what is the purpose of Table 9?

In graphs 4 and 5, we cannot compare different traits in the same graph because each scale is different (although CW and BW are similar and comparable). The trends can be standardized for a clearer comparison.

Author Response

We answered the following questions/comments:

Comment/question:

Usually, we call correlation or correlation coefficient but do not call heritability coefficient. We just call heritability or heritability estimate(s).

Answer:

We changed „coefficients of heritability“ or „heritability coefficient“ to „heritability“, see revised manuscript (lines 33, 34, 158, 326, 329, 342, 374, 393, 405, 464).

Comment/question:

Basically, CDG and CW are genetically the same trait. One of those traits can be removed. Other highly correlated traits (BW and MUS) can be also removed to reduce the evaluation cost if necessary.

Answer:

We agree with reviewer, we used all thede traits only for researsch purpose. These traits are removed in routine estimating of BVs.

Comment/question:

Lines 23: What is “resp.”? Please spell out and then define it.

Answer:

It means „respectively“. We changed the sentences with „resp.“ and removed this word, see revised manuscript (lines 22-25, 328).

Comment/question:

Lines 26-36: Heritability and correlations should be consistent by using the same number of decimals. 13.74% (=0.1374) and 33.41% (=0.3341) for heritability can be 14% (or 0.14) and 33% (or 0.33); 0.8351 and 0.8244 for correlation can be 0.84 and 0.82. Later, on the other hand, 0.17 – 0.20 and 0.79 – 0.97 were used.

Answer:

We uniformed coefficients of heritabilities and genetic parameters (see revised manuscript, lines 27, 28, 348-351(tab 3), 354, 360,  370-372 (tab 4), 386-378 (tab 5) and line 458).

Comment/question:

Line 32: Does this “A genetic heritability evaluation” mean “heritability estimation” or “genetic evaluation”?

Answer:

We corrected the sentence in manuscript. Correct term is: „genetic evaluation“ (line 33).

Comment/question:

Lines 89-90: What are meat yield (DP) and net gain (CDG)? Please define them.

Answer:

CDG is ratio of weight of carcass (def. of carcass, beef quarters: the muscle, bone and fat associated with the slaughter of an animal, left after the removal of the head, hide, legs and internal organs) to age of animal at slaughter. CDG = (weight of carcass in kg *1000) / age in days. DP is percentage of weight of carcass from live body weight before slaughtering (DP = (kg carcass/kg body weight)*100). We add definition to the revised manuscript (lines 95-98).

Comment/question:

Line 98: Is the grade (E to P) a subjective measurement by human or an objective by a measure?

Answer:

Grades E to P (the SEUROP system) is official system of subjective grading of carcasses of animals. It is not measuerement, only subjective assessment by qualified human on slaughterhouse.

Comment/question:

Line 155: “σg1 a σg2” are not variances.

Answer:

We corrected this mistake…see revised manuscript (lines 164-165).

Comment/question:

Lines 439-442: If meat traits are negatively correlated with milk, how can both traits be genetically improved? Discussion is needed before conclusion, by showing some evidence or citing other studies for the same breed.

Answer:

We would like to state, in chapter „Conclusion“, that there is general assuming (in dairy cattle public opinion) about negative interrelationship between milk and beef traits by cattle (in general). Also it is generally known positive relationship between milk yield and body size (body weight) of cows. So we assumed that generally there could be positive relationship through the body weight. We corrected the sentence in chapter Conclusion (lines 471-472).

Genetic improvement for milk and beef yield is possible using simultaneous selection (selection indices – like a Total merit index).

Comment/question:

I suggest not to use percentage in heritability as used in Tables 3, 4, and 5 and some sentences in the text. In results and discussion, most of the results were not expressed in percentage. Also, the number of decimals for heritability estimates and correlations should be consistent, using always 2 decimals (0.xx).

Answer:

We uniformed coefficients of heritabilities and genetic parameters and other numbering (see revised manuscript, lines 27, 28, 348-351(tab 3), 354, 360,  370-372 (tab 4), 386-378 (tab 5) and line 458).

Comment/question:

In Table 9, we cannot compare SD for S and F because those values depend on estimates of variance components for additive genetic and residual effects. Also, we cannot compare the means between S and F because the range of years is different. So, what is the purpose of Table 9?

Answer:

We removed Table 9 from manuscript (lines 480-409 and whole table 9 on the last page).

Comment/question:

In graphs 4 and 5, we cannot compare different traits in the same graph because each scale is different (although CW and BW are similar and comparable). The trends can be standardized for a clearer comparison.

Answer:

We changed graph 4 and 5 and change the text in chapter 3.4 (see lines 427-429, 435-445 and 450-453). We counted relative breeding values (estimated breeding values were standardized on mean 100 and standard deviation 12) for comparing trends in graphs (see in revised manuscript lines 435-438). Transformation on relative BV (RBV) according this formula:

RBV = ((BV – xBV)/sdBV*12) + 100

xBV is mean of all BV, sdBV is standard deviation of all BV.
